# Acute Colonic Diverticulitis: CT Findings, Classifications, and a Proposal of a Structured Reporting Template

**DOI:** 10.3390/diagnostics13243628

**Published:** 2023-12-08

**Authors:** Francesco Tiralongo, Stefano Di Pietro, Dario Milazzo, Sebastiano Galioto, Davide Giuseppe Castiglione, Corrado Ini’, Pietro Valerio Foti, Cristina Mosconi, Francesco Giurazza, Massimo Venturini, Guido Nicola Zanghi’, Stefano Palmucci, Antonio Basile

**Affiliations:** 1Radiology Unit 1, University Hospital Policlinico “G. Rodolico-San Marco”, 95123 Catania, Italy; davidegiuseppecastiglione@gmail.com (D.G.C.); corrado.ini@gmail.com (C.I.); 2Department of Medical Surgical Sciences and Advanced Technologies “GF Ingrassia”, University Hospital Policlinico “G. Rodolico-San Marco”, 95123 Catania, Italy; s.dipietro@studium.unict.it (S.D.P.); dario.milazzo2015@libero.it (D.M.); sebastianogalioto1996@gmail.com (S.G.); pietrovalerio.foti@unict.it (P.V.F.); spalmucci@sirm.org (S.P.); basile.antonello73@gmail.com (A.B.); 3Department of Radiology, IRCCS Azienda Ospedaliero-Universitaria di Bologna, Sant’Orsola-Malpighi Hospital, 40138 Bologna, Italy; cristina.mosconi@aosp.no.it; 4Interventional Radiology Department, Cardarelli Hospital of Naples, 80131 Naples, Italy; francescogiurazza@hotmail.it; 5Department of Diagnostic and Interventional Radiology, Circolo Hospital, Insubria University, 21100 Varese, Italy; venturini.massimo@hsr.it; 6Department of General Surgery, University of Catania, 95123 Catania, Italy; gzanghi@unict.it

**Keywords:** diverticulitis, colonic, computed tomography, structured report, abscess, fistula, perforation, peritonitis

## Abstract

Acute colonic diverticulitis (ACD) is the most common complication of diverticular disease and represents an abdominal emergency. It includes a variety of conditions, extending from localized diverticular inflammation to fecal peritonitis, hence the importance of an accurate diagnosis. Contrast-enhanced computed tomography (CE-CT) plays a pivotal role in the diagnosis due to its high sensitivity, specificity, accuracy, and interobserver agreement. In fact, CE-CT allows alternative diagnoses to be excluded, the inflamed diverticulum to be localized, and complications to be identified. Imaging findings have been reviewed, dividing them into bowel and extra-intestinal wall findings. Moreover, CE-CT allows staging of the disease; the most used classifications of ACD severity are Hinchey’s modified and WSES classifications. Differential diagnoses include colon carcinoma, epiploic appendagitis, ischemic colitis, appendicitis, infectious enterocolitis, and inflammatory bowel disease. We propose a structured reporting template to standardize the terminology and improve communication between specialists involved in patient care.

## 1. Introduction

Diverticular disease is one of the most common gastrointestinal disorders, especially in industrialized countries; it includes diverticulosis, defined as the presence of one or more diverticula that can develop in either the colon or small bowel, and diverticulitis, characterized by inflammation of the same [1].

Pseudodiverticula of the colon or false diverticula are defined as herniation of the mucous and submucous layers through a wall defect of the muscular layer of an intestinal loop between the taenia coli and the mesentery at the point of penetration of the blood vessel [2], often present in the sigmoid colon; true diverticula contain all layers of the intestinal wall, are congenital in etiology, and are usually detected in the ascending colon [3].

Pseudodiverticula genesis is multifactorial; it is believed that the low-fiber diet, the altered gut microbiome, and genetic factors are implicated in the development of diverticular disease; in particular, the hardening of the feces produces an increase in intraluminal pressure, which leads to herniation of the internal parietal layers through loci minoris resistentiae [3,4].

Acute colonic diverticulitis (ACD) represents the most common complication of diverticular disease since it occurs in about 10–25% of patients. ACD represents an abdominal emergency and encompasses a variety of conditions, ranging from localized diverticular inflammation to fecal peritonitis [2,5].

The pathophysiological basis of diverticulitis is thought to lie in ostial obstruction of the diverticulum by feces or food debris, leading to mucosal damage and subsequent ischemia, micro-perforation, and infection; however, new studies suggest that the state of chronic inflammation and alterations in the gut microbiome are involved in the pathogenic process [6].

Contrast-enhanced computed tomography (CE-CT) represents the diagnostic tool of first choice in patients with suspected diverticulitis [7,8,9,10], showing high diagnostic sensitivity (98–99%), specificity (99–100%), accuracy (98−99%), and excellent interobserver agreement [10,11,12].

CE-CT allows, in the scenario of suspected diverticulitis, to 1. confirm the diagnosis and identify the inflamed diverticulum; 2. stage the disease, stratifying patients for operative versus nonoperative treatment; 3. identify complications, depicting the extracolonic disease extent; 4. provide a valid tool in preoperative surgical and radiological interventional planning; and 5. suggest alternative diagnoses presenting similar clinical symptoms and signs (such as neoplasm, inflammatory bowel disease, appendicitis, epiploic appendagitis, and colon ischemia) [12,13].

Treatment of ACD depends on whether it is uncomplicated or complicated and the degree of complications. In the case of uncomplicated diverticulitis, the use of antibiotics is not recommended, and management of the patient in an outpatient setting is recommended. In complicated diverticulitis, in addition to broad-spectrum intravenous antibiotics and intestinal rest, the lower stages are treated with non-operative management, in which interventional radiology intervenes through percutaneous drainage of the abscesses [8,14].

In cases of peritonitis or severe complications, surgical management is necessary, especially in an emergency setting; options include colostomy formation, colonic resection with the construction of an end-colostomy (Hartmann procedure), and colonic resection with primary anastomosis with or without diverting loop ileostomy [14].

This article aims to describe the main CT findings of ACD using the cases in our database, illustrate the main classifications, and propose a structured report template for use in the emergency setting.

### CT Protocol

In our institution, in the clinical scenario of suspected ACD—with pain in the left lower quadrant, fever, and elevated inflammatory indexes—computed tomography (CT) examination is performed before and after intravenous administration of iodinated contrast medium. The image acquisition extends from the diaphragm to the pubic symphysis.

The technical parameters kV and mAs should be adjusted depending on the patient’s waist circumference to optimize the image quality and radiation dose.

The CT protocol consists of a non-contrast phase and a portal venous phase (acquired with a delay of 65–70 s). In the case of concomitant presence of clinical and laboratory signs of bleeding, an arterial phase (acquired with an automatic bolus tracking technique with the Region of Interest placed in the proximal abdominal aorta and a delay of 15 s) and a delayed phase were also acquired.

The CT examinations were performed using two different scanners: the Optima 660 (GE Healthcare, Chicago, IL, USA) and the Revolution EVO (GE Healthcare, Chicago, IL, USA).

A bolus of 80–120 mL of Iomeprol (Iomeron 350 or 400 mg/mL; Bracco Imaging, Milan, Italy), followed by a 30 mL saline flush, was administered using an automated contrast injection system at a flow rate of 3–4 mL/s. The CT images were reformatted in both the coronal and sagittal planes.

We believe that the use of oral contrast medium may not be indicated in the setting of ACD; the non-use does not compromise the effectiveness of the CT in the diagnosis [15], it is not recommended in clinical practice for long preparation times, and the large volume to be ingested [16] and studies show that it does not provide any diagnostic benefit [17].

Rectal administration of contrast medium may be considered in cases of suspected sinus tract from the rectosigmoid colon to an adjacent pelvic organ [3] or to evaluate for postoperative perforation or leakage [7].

## 2. CT Findings: What to Look for

Imaging findings in acute colonic diverticulitis can be divided into imaging findings related to the bowel wall and extra-intestinal wall signs.

### 2.1. Bowel Wall Imaging Findings

The primary finding of the intestinal wall in evaluating diverticulitis is the mural thickening (Figure 1). The intestinal wall, if distended, has a thickness ranging up to 3 mm [18]; if the segment of the colon is contracted, the wall thickness is considered normal up to 8 mm [19].

For the correct mural measurement, it is necessary to identify the longitudinal centerline of the colon, and measurements were made perpendicular to the centerline, excluding abscess and lumen, to identify the maximum distance from the serosal-to-mucosal surface of the colon, including the folds and teniae coli (Figure 1). When a lumen could not be seen, the entire serosa-to-serosa distance was measured and divided in half [20]. Dickerson et al. showed that maximum colonic wall thickness in patients with ACD predicts recurrent diverticulitis and may be helpful in stratifying patients according to the need for elective partial colectomy [20].

The length of the involved colon should also be measured and indicated based on the useful information provided with multiplanar reconstruction and 2D and 3D reformatting software [13]. Often, the mural thickening is associated with an identifiable inflamed diverticulum; in these cases, it is identifiable as an “arrowhead sign” if a portion of fluid crosses the edematous neck of the diverticulum [3]. A severity scale proposed by Dickerson et al. quantifies diverticula based on the distance from each other as few (more than 5 cm in between), mild (1–5 cm), moderate (<1 cm), and severe (no distance) [20].

### 2.2. Extra-Intestinal Wall Imaging Findings

#### 2.2.1. Mesenteric Findings

An increased density of pericolic fat [fat stranding] and a small amount of pericolic fluid represent the main mesenteric findings, which, combined with the mural thickening, suggest a localized inflammatory process in uncomplicated diverticulitis (Figure 2) [10,12,13]. The degree of fat stranding can range from “dirty fat” to peridiverticular phlegmon. The phlegmon consists of an inflammatory mass, without walls, located near the inflamed colonic tract, round or oval in shape, and on CT, it presents with high attenuation compared to the mesenteric fat, without an enhancing wall (Figure 3) [3,5]. Multiplanar reconstruction allows for identifying minimum amounts of pericolic fat stranding in the case of horizontal colonic segments [12]. Pereira et al. suggest that the presence of “disproportionate” fat stranding concerning mural thickening suggests the diagnosis of diverticulitis [21]. A small amount of fluid on the root of the mesentery [comma sign] and thickening of the lateroconal fascia are additional signs of an inflammatory process (Figure 1) [13].

#### 2.2.2. Vascular Findings

The inflammatory process could cause engorgement of the mesenteric vessels at the involved colonic tract, which is appreciated on CT images as the “centipede” sign [18]. ACD is one of the most common causes of pylephlebitis, ascending septic thrombophlebitis, characterized by inflammation and septic thrombosis of the mesenteric and portal venous systems [2,5]. Pylephlebitis represents the extension of the septic process to the efferent venous system from the inflamed bowel region [22]. CE-CT imaging shows filling defects in the mesenteric or portal vein, gas or soft tissue density within the vein (representing purulent material), and circumferential stranding of the perivascular fat can be appreciated [3]. Complications of thrombophlebitis include the development of liver abscesses, septic emboli, venous rupture, and pulmonary thromboembolism [23]. Representing the diverticulum as an outpouching from a parietal defect from where the blood vessels penetrate, diverticular bleeding can be an occurrence, more appreciable in chronic diverticulitis [3,24]. On unenhanced CT, hyperdense intraluminal contents can suggest bleeding; after contrast medium, the presence of active contrast extravasation with enlarging contrast volume on the portal venous phase represents the finding of active bleeding (Figure 4).

#### 2.2.3. Findings of Complications

CT findings of acute complicated diverticulitis include extra-luminal free-air, abscess, intra-abdominal free-fluid, fistula, and bleeding [12].

The severe inflammatory state of the colonic wall could lead to mural necrosis and perforation of the bowel [25]. Free air can be well-contained and self-limiting, consisting of localized gas adjacent to the inflamed colon (air bubbles or pockets), or it can spread into the intraperitoneal or retroperitoneal cavity [5,8]. Pericolic free air is defined as the presence of air bubbles or air collection within 5 cm of the inflamed bowel segment without distant air (Figure 5), or distant free air collections in the abdominal or retroperitoneal cavity with a distance >5 cm from the inflamed bowel segment (Figure 6) [11]. CT can detect signs of perforation, including bowel wall discontinuity and extraluminal air, especially using lung window settings (Figure 6) [26].

Diverticulitis can cause abscess formation. An abscess is defined as a fluid-containing mass with or without air and an avidly enhancing wall that can be intramural, pericolic, or distant from the inflamed colon (Figure 6 and Figure 7). It is necessary to report the inflamed colonic tract’s localization, the abscess’s maximum diameter (<4 cm or ≥4 cm), and the anatomical relationships with the surrounding structures, especially intestinal loops or organs, with which fistulas can develop. Regarding the dimensions, a diameter greater than or equal to 4 cm indicates percutaneous drainage (Figure 7) [8,10].

The most common sites of distant abscesses are the liver, adnexa, lungs, brain, and spine [5,27]. The tubo-ovarian abscess can be caused by the close contiguity of the sigmoid colon with the adnexa; instead, colonic mucosal defects can cause the hematogenous spread of bacteria and induce the formation of abscesses in distant sites. The presence of an abscess in direct contact with another anatomical structure, violating its parietal integrity, leads to the formation of a fistula [25]. Fistulas occur in about 2% of cases of acute diverticulitis [3], and the most common types, in descending order, are colovesical, colocutaneous, colovaginal, coloenteric, and colouterine [28]. Colovesical fistulas are more frequent in men due to the protective effect of the uterus in women [29]. It manifests clinically as dysuria, pneumaturia, or fecaluria, localized above the left posterior side of the bladder due to its proximity to the sigmoid colon [16]. The CT shows the lack of a fatty cleavage plane between the colon and the bladder, the thickening of the organ wall, and the presence of free intravesical air (Figure 8) [30]. Administration of contrast medium into the bladder or rectum and fluid-sensitive sequences in magnetic resonance imaging may help locate the sinus tract [3,5].

The presence of air bubbles in the uterine cavity could be a sign of a myometrial abscess due to the formation of a colouterine fistula. In contrast, in hysterectomized patients, the colovaginal fistula should be excluded [22].

Peritonitis is a life-threatening complication with visceral perforation as its etiological cause. Peritonitis can be purulent when the perforation concerns a pericolic abscess or feculent [stercoraceous] when the perforation concerns a non-inflamed diverticulum [3,31]. On CT, the signs of peritonitis are represented by free fluid in the peritoneal cavity and thickening and enhancement of the peritoneal layers (Figure 9); Sartelli et al. defined diffuse fluid as the presence of abdominal effusion in at least two distant quadrants [10].

## 3. Staging and Management

In 1978, Hinchey et al. classified the severity of ACD into four stages, based on intraoperative findings, to guide surgical management [31].

An exclusively peri-colonic phlegmonous change or abscess characterizes stage I, whereas a distant pelvic, retroperitoneal, or intra-abdominal abscess categorizes stage II. Stage III includes purulent peritonitis, while stage IV occurs when a large perforation of the loop causes fecal or stercoraceous peritonitis.

With the advent of CT, the Hinchey classification has changed to integrate the radiological findings; in 2005, Kaiser et al. [32] modified the Hinchey classification based on CT findings. The modified Hinchey classification included stage 0 (mild clinical diverticulitis), in which the presence of a diverticulum is associated or not with colonic wall thickening; stage Ia (confined pericolonic inflammation/phlegmon), in which colonic wall thickening is associated with pericolic soft tissue changes; and stage Ib (pericolonic/mesocolic abscess), in which the pericolonic/mesocolic abscess appears [32].

To overcome the lack of categorization on chronic changes, Schreyer et al. developed a classification (Table 1) in which they include chronic diverticular disease, recurrent or chronic symptomatic diverticular disease (type 3), and diverticular bleeding (type 4).

The American Association for the Surgery of Trauma (AAST) developed a grading scale from I (mild disease) to V (severe disease) for ACD that addresses clinical, radiologic, operative, and pathologic grades of disease [33].

Some pilot studies showed that, compared with the modified Hinchey classification, the AAST grade better predicted the decision to operate [34] and was equivalent in predicting procedural intervention and complications [35].

In 2015, a new classification based on radiological findings was proposed and included in the WSES guidelines for managing ACD [8,10]. This classification divides ACD into two groups: complicated and uncomplicated. Uncomplicated ACD is characterized only by thickening of the wall of the diverticula, with increased density of the peri-colic fat. Complicated diverticulitis is divided into the Ia stage with pericolic air bubbles or little pericolic fluid without abscess; Ib with abscess ≤4 cm; IIa with abscess >4 cm; IIb with distant air; III with diffuse fluid without distant free air; and IV with diffuse fluid with distant free air.

At each stage, more appropriate management is suggested, from conservative treatment with broad-spectrum antibiotic therapy (stages Ib and Ia) to percutaneous drainage (stage IIa), up to surgical treatment, from resection with anastomosis (stage IIb or III in stable patients) to Hartmann’s resection (stage IIb or III in unstable patients or stage IV) [8].

A recent multicenter study comparing the three most used classifications showed that the AAST, WSES, and modified Hinchey classifications were similar in predicting complications, reintervention, and mortality rates [36]. The area under the curve (AUC) for the need for surgery and the occurrence of major complications was higher for AAST and modified Hinchey scores [36].

## 4. Differential Diagnosis

The main differential diagnoses of acute colonic diverticulitis include colon carcinoma, epiploic appendagitis, ischemic colitis, appendicitis, infectious enterocolitis, and inflammatory bowel disease [2,12]. Colon adenocarcinoma must be excluded from the differential diagnoses, as it presents as an eccentric or circumferential mural thickening [12]. Compared to ACD, the colonic segment is <5–10 cm involved, there are no diverticula, and there is an abrupt transition between the thickened segment and the normal loop (shoulder sign) (Figure 10) [2,12]. Local lymphadenopathy suggests colorectal cancer [37].

The epiploic appendages are protrusions of fatty tissue located on the antimesenteric side of the colon. In case of torsion or thrombosis of their vascular pedicle, they can manifest ischemia and inflammation and aim for acute diverticulitis with phlegmon [38]. CE-CT shows an oval mass with adipose tissue density located on the antimesenteric side and with peripheral hyperattenuating (hyperattenuating ring sign), expression of inflamed fat, and a central focal area of hyperattenuating (central dot sign), which corresponds to the thrombosed vein within the inflamed appendage (Figure 11) [39]. Compared with acute diverticulitis, the associated diverticular disease is not present in appendagitis without colonic wall thickening.

## 5. Proposed Reporting Template

Despite different classifications for diagnosing and managing ADC, to our knowledge, no reporting templates have been proposed for the structured reporting of acute diverticulitis and its complications.

Some studies demonstrated how greater standardization in reporting leads to improved communication and more complete reports and reduces variability and error rates [40,41].

Structured reporting facilitates the radiology resident’s learning, providing a systematic approach to recognizing the specific pathology’s key characteristics to be reported in the radiology report [42].

Structured reporting has been introduced in other abdominal pathologies, particularly in oncology abdominal imaging, such as rectal cancer, where it has become necessary in staging and restaging and facilitates clinical decision making [43].

The proposed reporting template concerns acute diverticulitis and is designed to be used with contrast-enhanced CT imaging, as shown in Figure 12.

This template incorporates the classification scheme and terminology of the WSES classification [8] while maintaining the flexibility to add free text for the qualitative aspects of the report. The model includes colonic wall findings and complication findings, divided into signs of perforation, abscess, free fluid, and vascular complications.

Our aim is to standardize the language, avoid the nuances expected by radiologists in their reports, improve communication between specialists involved in patient care, and standardize the multidisciplinary approach.

## 6. Conclusions

CE-CT is the diagnostic modality of choice if ACD is suspected. In fact, it plays an essential role in evaluating the extent, severity, and complications of this pathology and, therefore, allows the correct management and surgical strategy to be established.

In this paper, we have reviewed the most used classifications of ACD, proposing a new structured reporting template that, through a system of appropriate terminology, can reduce discrepancies between radiological reports and standardize the language among specialists in managing this condition.

Further studies are needed to validate improved outcomes in ACD patients using a standardized reporting template.

## Figures and Tables

**Figure 1 diagnostics-13-03628-f001:**
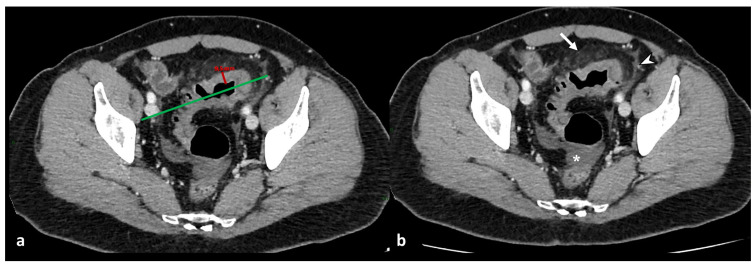
Axial contrast-enhanced CT images show acute left colonic diverticulitis and associated findings of perivisceritis (**a**,**b**). Once the longitudinal axis of the colon is identified (green straight line in (**a**)), bowel wall thickness is measured perpendicular to the centerline (9.6 mm, red segment in (**a**)), showing an increase in the maximum distance from the serosal-to-mucosal surface. Additional mesenteric findings (**b**), such as increased density of pericolic fat (arrow), thickening of latero-conal fascia (“comma sign”) (arrowhead), and abdominal free fluid (asterisk), are found.

**Figure 2 diagnostics-13-03628-f002:**
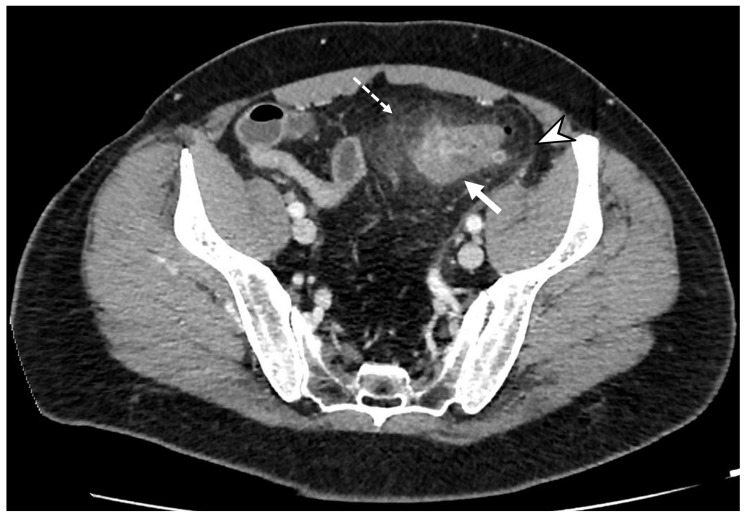
Axial contrast-enhanced CT images show wall thickening of the left colon (thick arrow) and the presence of perivisceritis, including the increased density of pericolic fat (fat stranding) (dotted arrow) and thickening of the left lateroconal fascia (so-called comma sign) (arrowhead).

**Figure 3 diagnostics-13-03628-f003:**
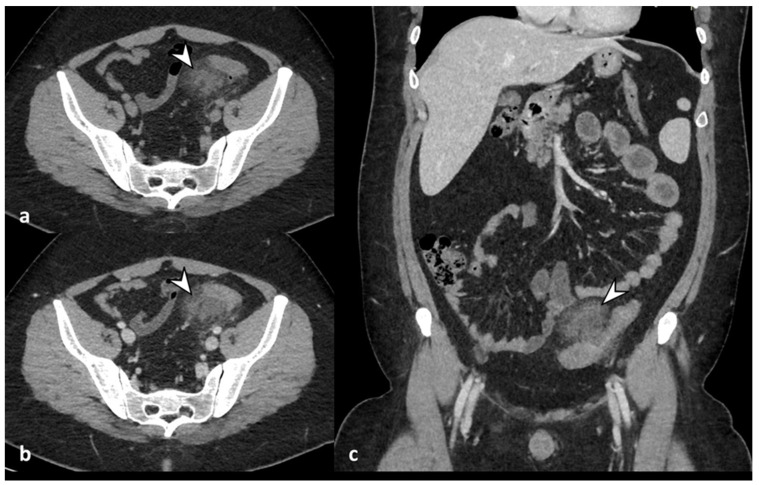
Axial unenhanced (**a**), axial and coronal contrast-enhanced CT images (**b**,**c**) show wall thickening of the left colon and inflammatory mass, without walls, located near the inflamed colonic tract, oval in shape, without an enhancing wall, that represents a peridiverticular phlegmon (arrowheads).

**Figure 4 diagnostics-13-03628-f004:**
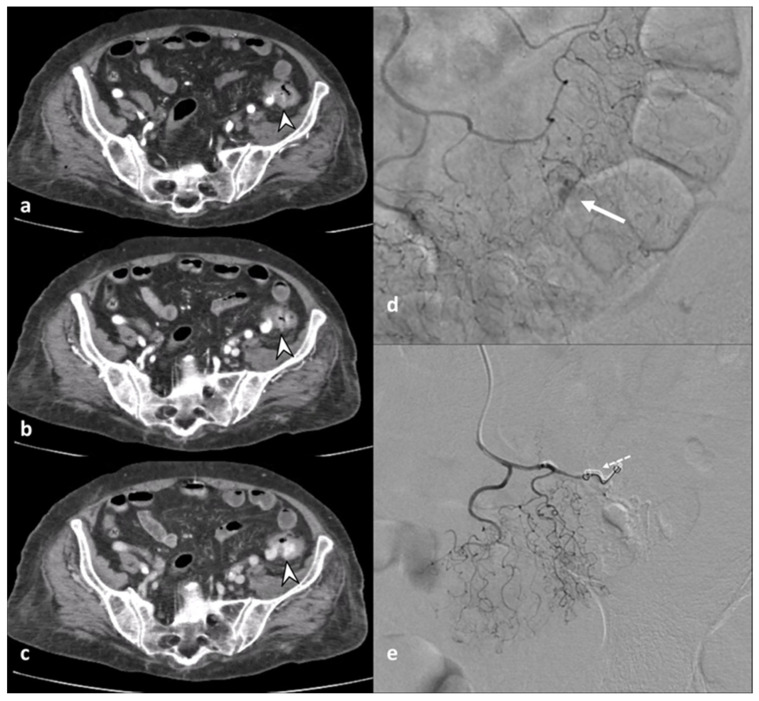
Axial contrast-enhanced arterial (**a**), portal (**b**), and delayed (**c**) CT phases show the presence of arterial active contrast extravasation within a diverticulum (arrowheads), more appreciable in the later phases of the study (**b**,**c**); this is a typical finding of active bleeding. Following DSA (**d**,**e**) reveals active extravasation of iodinated contrast medium (arrow), allowing identification of the source of the bleeding, which was then treated with coil embolization (dotted arrow in (**e**)).

**Figure 5 diagnostics-13-03628-f005:**
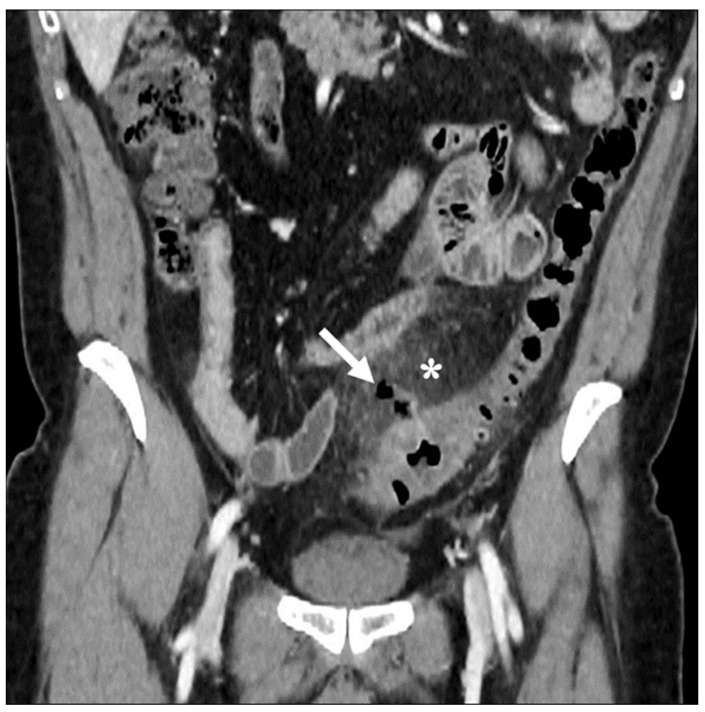
Coronal contrast-enhanced CT images show wall thickening of the left colon with signs of perivisceritis (asterisk) and localized air bubbles adjacent to the inflamed colon (arrow).

**Figure 6 diagnostics-13-03628-f006:**
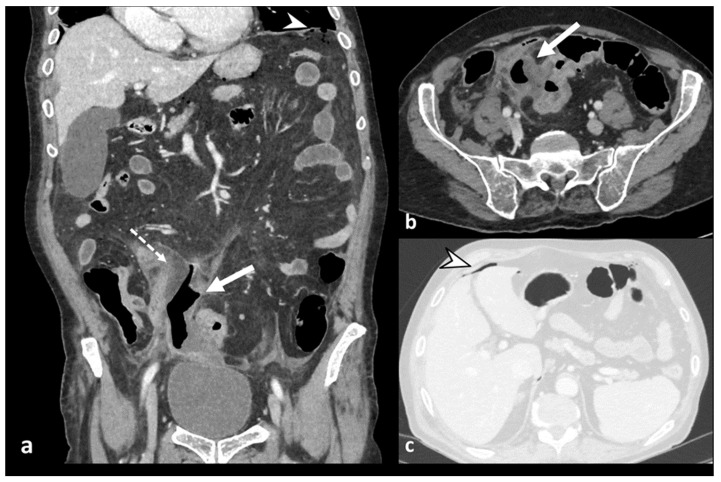
Axial and coronal contrast-enhanced CT images (**a**,**b**) show an abscess, a fluid-containing mass with air and an enhancing wall (white arrow), near the inflamed colonic tract (dotted arrow) and signs of perforation, with distant air bubbles below the left hemidiaphragm (arrowhead in (**a**)); lung window axial CT image (**c**) shows distant free air anteriorly to the liver (arrowhead).

**Figure 7 diagnostics-13-03628-f007:**
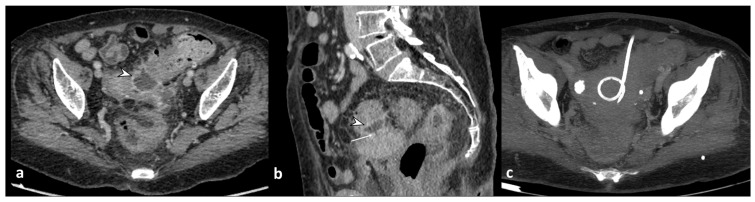
Axial (**a**) and coronal (**b**) contrast-enhanced CT images show an abscess near the sigmoid colon (arrowhead), located close to the uterine fundus (arrow). An axial unenhanced CT image (**c**) shows a 10 F pigtail catheter in the abscess collection.

**Figure 8 diagnostics-13-03628-f008:**
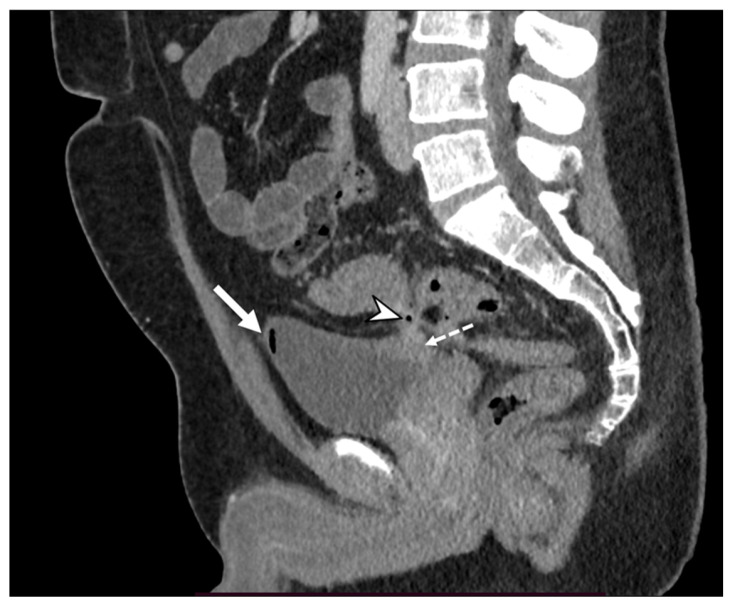
A sagittal contrast-enhanced CT image shows a fistulous tract (arrowhead) between the thickened wall of the sigmoid colon and the posterior bladder wall that appears thickened (dotted arrow) and the presence of free intravesical air (arrow). These findings are suggestive of a colovescical fistula.

**Figure 9 diagnostics-13-03628-f009:**
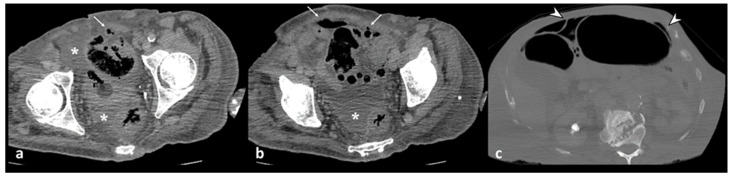
Axial contrast-enhanced CT images (**a**,**b**) show the presence of pericolic air bubbles and air collections (arrows) and diffuse peritoneal free fluid (asterisks); lung window axial CT (**c**) shows distant free air (arrowheads) below the abdominal wall.

**Figure 10 diagnostics-13-03628-f010:**
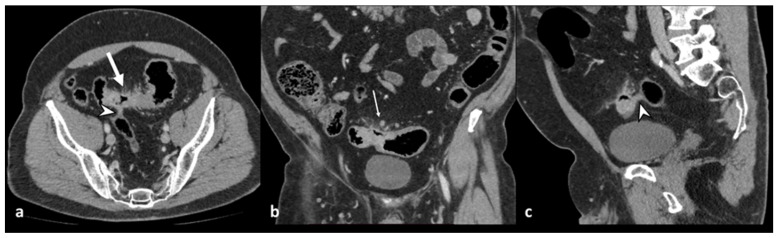
Axial (**a**), coronal (**b**), and sagittal (**c**) contrast-enhanced CT show stenosing annular colon adenocarcinoma (thick arrow in (**a**)), with its typical “apple core” appearance and associated malignant lymphadenopathy (thin arrow in (**b**)). There is also the presence of a colocolic fistula (arrowheads).

**Figure 11 diagnostics-13-03628-f011:**
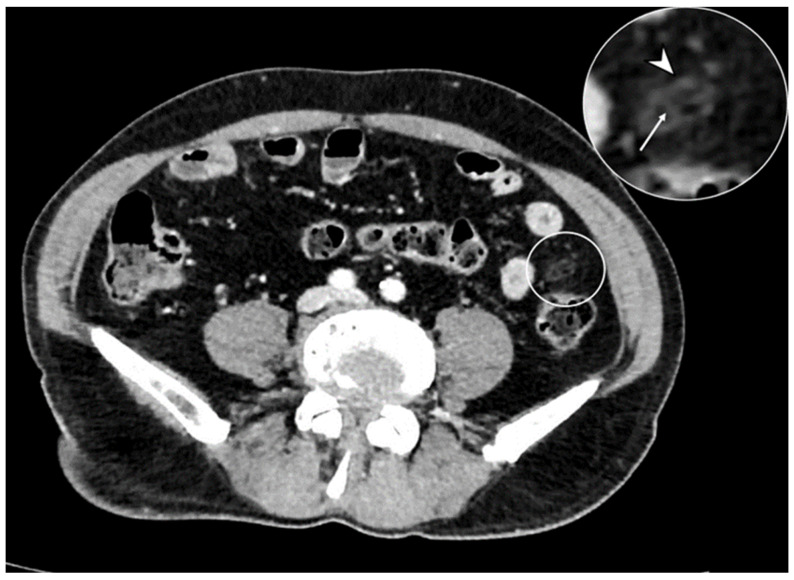
Axial contrast-enhanced CT shows an oval mass with adipose tissue density located on the antimesenteric side (circle), which presents fat inflammation, demonstrated by peripheral hyperdensity (arrowhead in detail) and central thrombosed vein (arrow in detail). These findings are consistent with epiploic appendagitis.

**Figure 12 diagnostics-13-03628-f012:**
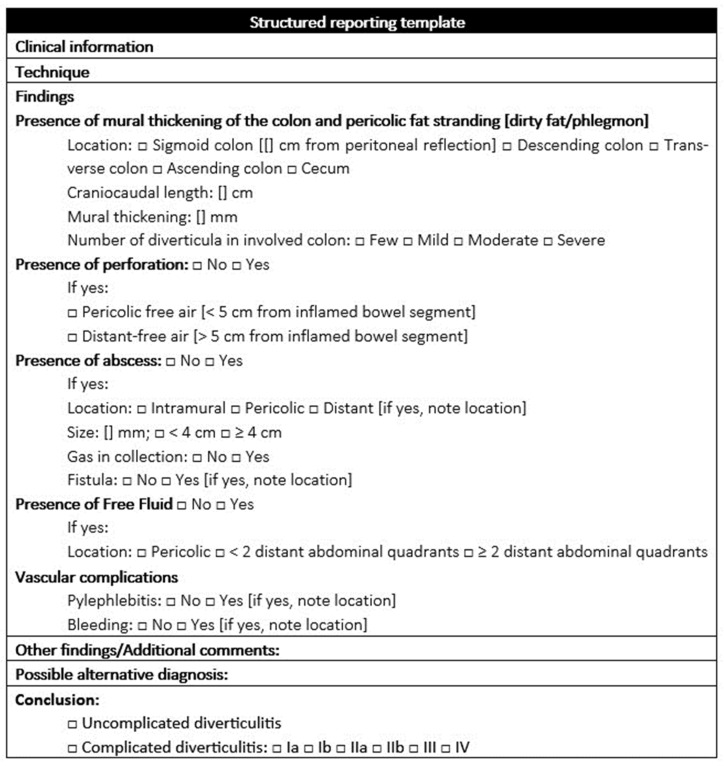
Structured reporting template.

**Table 1 diagnostics-13-03628-t001:** Classifications in use for ACD.

	Hinchey Classification [31]	Modified Hinchey Classification by Kaiser et al. [32]	CDD * by Schreyer et al. [24]	AAST ** Grade [33]	WSES Classification by Sartelli et al. [10]
**0**		Mild clinical diverticulitis	Asymptomatic diverticulosis		Uncomplicated diverticulitis
**1**	Pericolic abscess or phlegmon		Acute uncomplicated diverticular disease/diverticulitis	Colonic inflammation	
*1a*		Confined pericolic inflammation/phlegmon	Diverticulitis/diverticular disease without reaction of surrounding tissue		Pericolic air bubbles or little pericolic fluid without abscess
*1b*		Pericolonic/mesocolic abscess	diverticulitis with phlegmonous reaction of surrounding tissue		Abscess ≤ 4 cm
**2**	Pelvic, distant intraabdominal, or retroperitoneal abscess	Pelvic, distant intraabdominal, or retroperitoneal abscess	Acute complicated diverticulitis as in 1b, additionally	Colon microperforation or pericolic phlegmon without abscess	
*2a*			Microabscess		Abscess > 4 cm
*2b*			Macroabscess		Distant air (>5 cm from inflamed bowel segment)
*2c*			Free perforation (2c1 purulent peritonitis; 2c2 fecal peritonitis)		
**3**	Generalized purulent peritonitis	Generalized purulent peritonitis	Chronic diverticular disease recurrent or chronic symptomatic diverticular disease	Localized pericolic abscess	Diffuse fluid without distant free air (no hole in colon)
*3a*			Symptomatic uncomplicated diverticular disease (SUDD)		
*3b*			Recurrent diverticulitis without complications		
*3c*			Recurrent diverticulitis with complications		
**4**	Generalized fecal peritonitis	Generalized fecal peritonitis	Diverticular bleeding	Distant and/or multiple abscesses	Diffuse fluid with distant free air (persistent hole in the colon)
**5**				Free colonic perforation with generalized peritonitis	

* CDD—classification of diverticular disease; ** AAST—American Association for the Surgery of Trauma.

## Data Availability

Not applicable.

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
