# Peer review of "Acute Colonic Diverticulitis: CT Findings, Classifications, and a Proposal of a Structured Reporting Template"

_diagnostics, 2023, doi:10.3390/diagnostics13243628_

Round 1

Reviewer 1 Report

Comments and Suggestions for Authors

Summary:

Acute colonic diverticulitis (ACD) is a common complication of diverticular disease, ranging from localized inflammation to more severe conditions, including fecal peritonitis. An accurate diagnosis of ACD is essential, and contrast-enhanced computed tomography (CE-CT) plays a vital role due to its high sensitivity and specificity. CE-CT not only allows for the exclusion of alternative diagnoses, the localization of inflamed diverticula, and the identification of complications but also facilitates disease staging using classifications like Hinchey's modified and WSES. The authors also propose a reporting template to standardize terminology and enhance communication.

General concept comments:

The study is well-conducted and summarizes the most important findings in ACD that every radiologist should know and describe. The introduction provides ample background information. The choice of images is generally appropriate. The aim is clear, though the section about structured reporting is relatively brief compared to the theoretical aspects. In my opinion, there should be more emphasis on the template. 
ACD is a common disease, and the most critical details for a radiologist to describe are whether there is free perforation (yes/no) and the presence of an abscess (yes/no). In the “emergency setting,” where this disease is most commonly encountered, it might be time-consuming to complete this form, especially considering that radiologists still need to write reports on all other intraabdominal organs/findings.

Specific comments:

Line 31: You may want to specifiy "Pseudodiverticula of the colon" since there exist various locations for pseudodiverticula (including blood vessels).

Line 59 and 60: You could also call these inflammations "appendicitis" and "epiploic appendagitis?

Line 75-94: Why do you perform a "triple-phase" CT? In cases with suspected ACD, a single portalvenous phase would be enough, wouldn't it? Do you always want to exclude active bleeding even if there is no clinical suspicion? Which scanner and contrast medium did you use? 

Line 145: (c) turned into a copyright sign.

Line 187-190: Figure 6: I do not see a clear abscess formation marked with the dotted arrow (this could just be a phlegmon, maybe you have a better picture, or maybe in an axial plane). The solid arrows are said to point at the inflamed colon. Or are the arrows switched and the solid ones are pointing to the abscess formation?

Line 198-201: Figure 7: very nice placement of a drainage, not the easiest one!

Comments on the Quality of English Language

Moderate improvements are needed in the English language and grammar. For example, in line 16, remove "Indeed,"; in line 19, change "CE-CT allows to stage of the disease;" to "CE-CT allows staging of the disease"; and in line 179, consider revising "free air distant air collections." I highly recommend having your text proofread by a native English speaker.

Author Response

REVIEWER 1

General concept comments:

The study is well-conducted and summarizes the most important findings in ACD that every radiologist should know and describe. The introduction provides ample background information. The choice of images is generally appropriate. The aim is clear, though the section about structured reporting is relatively brief compared to the theoretical aspects. In my opinion, there should be more emphasis on the template.

ACD is a common disease, and the most critical details for a radiologist to describe are whether there is free perforation (yes/no) and the presence of an abscess (yes/no). In the “emergency setting,” where this disease is most commonly encountered, it might be time-consuming to complete this form, especially considering that radiologists still need to write reports on all other intraabdominal organs/findings.

RESPONSE: We thank the Reviewer for your consideration and for the positive comments on our manuscript.

The template section is short, but we think it is essential to give relevance to the theoretical aspects explained in the other sections, which constitute the basis on which the template was structured and whose understanding and knowledge is fundamental, in our opinion, to understand the organization and the usefulness of the template. Furthermore, several concepts have been implied in the template section because they have already been illustrated in the other sections.

We agree with the reviewer that the use of the template, as in many cases of structured reports, may be time-consuming at the beginning and therefore not suitable for initial use in the emergency, at least in the learning phase. But once radiologists become familiar with structured report and use it often, as experience suggests, the time required is significantly reduced and this makes it suitable even in emergency setting.

Specific comments:

 Line 31: You may want to specifiy "Pseudodiverticula of the colon" since there exist various locations for pseudodiverticula (including blood vessels).

RESPONSE: We thank the Reviewer for your observation. We made the suggested change.

Line 59 and 60: You could also call these inflammations "appendicitis" and "epiploic appendagitis?

RESPONSE: We thank the Reviewer for your observation. We made the suggested change.

 Line 75-94: Why do you perform a "triple-phase" CT? In cases with suspected ACD, a single portalvenous phase would be enough, wouldn't it? Do you always want to exclude active bleeding even if there is no clinical suspicion? Which scanner and contrast medium did you use?

RESPONSE: We thank the Reviewer for your observation, and we agree with that. We modified the manuscript specifying that arterial and delayed phase are acquired in the case of concomitant presence of clinical and laboratory signs of bleeding.

Line 145: (c) turned into a copyright sign.

RESPONSE: We thank the Reviewer for your observation. We corrected this typo.

Line 187-190: Figure 6: I do not see a clear abscess formation marked with the dotted arrow (this could just be a phlegmon, maybe you have a better picture, or maybe in an axial plane). The solid arrows are said to point at the inflamed colon. Or are the arrows switched and the solid ones are pointing to the abscess formation?

RESPONSE: We thank the Reviewer for your observation. We corrected this typo.

 Line 198-201: Figure 7: very nice placement of a drainage, not the easiest one!

RESPONSE: We thank the Reviewer.

Comments on the Quality of English Language

Moderate improvements are needed in the English language and grammar. For example, in line 16, remove "Indeed,"; in line 19, change "CE-CT allows to stage of the disease;" to "CE-CT allows staging of the disease"; and in line 179, consider revising "free air distant air collections." I highly recommend having your text proofread by a native English speaker.

RESPONSE: We thank the Reviewer for your observations, we made the suggested changes.

Reviewer 2 Report

Comments and Suggestions for Authors

This is a well-written manuscript on CT-scanning in acute colonic diverticulitis with the proposal of for a structured reporting template. The subject is relevant, and I have only some few comments. 

Introduction

In the introduction is stated that the treatment of uncomplicated diverticulitis is antibiotics and dietary restrictions. There is no evidence for this treatment, and it has been abandoned by most hospitals. In addition, there is no reason for hospitalization in these cases. Antibiotic treatment is controversial for the complicated diverticulitis with minor local abscess or local perforation, but there is no evidence at all for bowel-rest. 

CT protocol

Most departments will in addition to typical history require an increased CRP for the indication og CT-scan. 

The reporting template

Number of diverticula is too difficult to handle, and it has no clinical relevance. Suggest that it is deleted or simplified as 1) few (define the max number) or 2) several.

Author Response

REVIEWER 2

This is a well-written manuscript on CT-scanning in acute colonic diverticulitis with the proposal of for a structured reporting template. The subject is relevant, and I have only some few comments.

RESPONSE: We thank the Reviewer for your consideration and for the positive comments on our manuscript.

Introduction

In the introduction is stated that the treatment of uncomplicated diverticulitis is antibiotics and dietary restrictions. There is no evidence for this treatment, and it has been abandoned by most hospitals. In addition, there is no reason for hospitalization in these cases. Antibiotic treatment is controversial for the complicated diverticulitis with minor local abscess or local perforation, but there is no evidence at all for bowel-rest.

RESPONSE: We thank the Reviewer for your comments, we made the suggested changes.

CT protocol

Most departments will in addition to typical history require an increased CRP for the indication og CT-scan.

RESPONSE: We thank the Reviewer for your observations, as you suggest, the clinical suspicion of ACD and eventually the subsequent indication of CT scan typically requires the concordance of clinical data, signs, symptoms and laboratory data, including increased CRP.

The reporting template

Number of diverticula is too difficult to handle, and it has no clinical relevance. Suggest that it is deleted or simplified as 1) few (define the max number) or 2) several.

RESPONSE: We thank the Reviewer for your observations; we used a severity scale proposed by Dickerson et al., and inserted into the text (2.1 Bowel Wall Imaging Findings), that visually quantifies diverticula based on the distance from each other as few (more than 5 cm in between), mild (1-5 cm), moderate (< 1 cm), and severe (no distance).

Reviewer 3 Report

Comments and Suggestions for Authors

Dear authors,

Many thanks for providing me the opportunity to review your excellent article on acute colonic diverticulitis. It is very well written and a great overview. 

My only remark would be to slightly alter the organization of the content using less new paragraphs, since often the topic continues. There are a few further minor suggestions that you will see in the attached PDF for your consideration.

Many thanks!  

Comments on the Quality of English Language

Excellent use of English, with a few minor suggestions as included in the attached PDF

Author Response

REVIEWER 3

Many thanks for providing me the opportunity to review your excellent article on acute colonic diverticulitis. It is very well written and a great overview. 

My only remark would be to slightly alter the organization of the content using less new paragraphs, since often the topic continues. There are a few further minor suggestions that you will see in the attached PDF for your consideration.

RESPONSE: We thank the Reviewer for your consideration and the positive comments on our manuscript. We made all the suggested changes.